# Fabrication of Laser-Induced Graphene Based Flexible Sensors Using 355 nm Ultraviolet Laser and Their Application in Human–Computer Interaction System

**DOI:** 10.3390/ma16216938

**Published:** 2023-10-29

**Authors:** Binghua Sun, Qixun Zhang, Xin Liu, You Zhai, Chenchen Gao, Zhongyuan Zhang

**Affiliations:** 1Key Laboratory of CNC Equipment Reliability, Ministry of Education, School of Mechanical and Aerospace Engineering, Jilin University, Changchun 130025, China; 2Chongqing Research Institute, Jilin University, Chongqing 401100, China; 3Institute of Structured and Architected Materials, Liaoning Academy of Materials, Shenyang 110167, China; 4College of Automotive Engineering, Jilin University, Changchun 130025, China

**Keywords:** laser-induced graphene, commercial PI film, flexible sensors, human–computer interaction system, 355 nm ultraviolet laser

## Abstract

In recent years, flexible sensors based on laser-induced graphene (LIG) have played an important role in areas such as smart healthcare, smart skin, and wearable devices. This paper presents the fabrication of flexible sensors based on LIG technology and their applications in human–computer interaction (HCI) systems. Firstly, LIG with a sheet resistance as low as 4.5 Ω per square was generated through direct laser interaction with commercial polyimide (PI) film. The flexible sensors were then fabricated through a one-step method using the as-prepared LIG. The applications of the flexible sensors were demonstrated by an HCI system, which was fabricated through the integration of the flexible sensors and a flexible glove. The as-prepared HCI system could detect the bending motions of different fingers and translate them into the movements of the mouse on the computer screen. At the end of the paper, a demonstration of the HCI system is presented in which words were typed on a computer screen through the bending motion of the fingers. The newly designed LIG-based flexible HCI system can be used by persons with limited mobility to control a virtual keyboard or mouse pointer, thus enhancing their accessibility and independence in the digital realm.

## 1. Introduction

Human–computer interaction (HCI) technology refers to the design, development, and study of interfaces between humans and computers [1,2]. Researchers have been working on HCI systems toward a more wearable, multifunctional, and intelligent design [3]. Besides traditional HCI devices, such as keyboards, mouses, and touchscreens, novel HCI devices based on Voice Recognition [4], Gesture Recognition [5], Eye Tracking [6], brain–computer interfaces [7], etc., have been developed in the past decades. Recently, wearable devices, especially hand gesture-based flexible wearable devices [8,9], have greatly influenced the field of HCI by introducing new ways for users to interact with technology. These devices are worn on the body and provide convenient access to digital functionalities and data. HCI systems [10] based on wearable devices are probably the most natural, comfortable, and intuitive way for humans to interact with machines because they are very close to the way humans interact with each other.

Sensors play a crucial role in enabling the functionality and capabilities of flexible wearable devices [11,12]. As the core component of flexible wearable devices, sensors are available in inflexible [13] and flexible [14] forms. The materials used to make inflexible sensors are silicon [15,16] and lead zirconate titanate (PZT) [17,18]. While rigid sensors can provide accurate sensing capabilities, there are some disadvantages when incorporating them into flexible wearable devices, such as limited flexibility, potential for damage, design and integration challenges, and especially comfort and user experiences [19,20]. Addressing these drawbacks is crucial for improving the comfort, flexibility, and overall user experience of flexible wearable devices [21].

Compared with rigid sensors, flexible sensors are more comfortable and natural, as they can conform to the wearer’s movements [22,23]. Additionally, they are simple and inexpensive to manufacture [24] compared with rigid sensors. The past decade has seen the development of materials, bendable and stretchable sensors, manufacturing techniques, etc., in flexible sensor-based wearable devices. In 2014, Jong, s et al. developed a new method to prepare flexible pressure sensors based on pressure-sensitive rubber [25]. In 2019, Xu et al. fabricated a fluid soft sensor using a conductive solution (potassium iodide and glycerol solution) [26]. In the same year, Takada et al. introduced a data glove based on conductive optical fiber [27], and the bending movement of the fingers could be detected as the resistance of the conductive fiber decreased when the fingers bent. J.Y. Huang et al. fabricated a strain sensor based on a thermoplastic polyurethane electrospun membrane (TPUEM) bridged with carbon nanotubes (CNT) and silver nanoparticles (AgNPs) [28]. This sensor showed high sensitivity and a large strain range and strain coefficient. While flexible sensors have made significant progress, there are still some limitations and disadvantages associated with their use in flexible wearable devices, such as accuracy and sensitivity, calibration and drift, limited lifespan complexity of integration, etc.

In 2014, Tour et al. first reported the fabrication of laser-induced graphene (LIG) on a commercial polyimide (PI) film under ambient conditions using a CO_2_ laser system [29], which opened up new avenues for the applications of flexible wearable sensors [30]. Due to the remarkable properties [29] (flexibility, high porosity (340 m^2^g^−1^), high thermal stability (900 °C), and good electrical conductivity (25 S cm^−1^)) and versatile fabrication process (fast speed, non-mask, and low-cost customizable preparation) [31,32,33], LIG enables the fabrication of high-performance flexible sensors [34], which could be used in flexible wearable devices that require high flexibility and versatility [35]. Qing Tian et al. proposed an ultrasensitive self-healing pressure sensor based on a pea pod with excellent stability (more than 1000 times the load cycles) and applied it to the detection of pulse rate [36]. Inspired by fingerprints, resistive strain sensors with fast response, balanced sensitivity and strain range (gauge factor: 42–50%, strain range: 191.55), and good stability (>1500 cycles) were prepared by Wentao Wang et al. [37]. Various human body movements were detected using this resistive strain sensor. Chenghan Yi et al. showed a flexible piezoresistive pressure sensor with excellent (performance) long-term cycle stability (>1800 cycles) and zero standby power consumption (bending angle: 0–5°) [38]. With this sensor, they successfully equipped a bioinspired artificial tactile neuron. In summary, laser-induced graphene [39] offers a range of exciting possibilities for the development of flexible wearable devices. Its unique combination of flexibility [40], lightweight nature, conductivity, and sensing capabilities [41] makes it a promising material for the next generation of wearable technology [42,43].

The fabrication of flexible sensors based on LIG technology and their applications in human–computer interaction (HCI) systems are presented in this paper. LIG was generated through direct laser interaction with a commercial polyimide (PI) film, and the effect of laser parameters (laser power, scanning speed, processing numbers, and scanning intervals) on the properties of LIG was investigated. Furthermore, flexible sensors were fabricated through a one-step method using the as-prepared LIG under the optimal laser parameters, and the performance of the flexible sensor was evaluated through tensile and bending tests. In addition, a flexible wearable glove was fabricated through the integration of the flexible sensors, and a flexible glove and its application in the HCI system was demonstrated. The newly designed flexible HCI system forms a bridge for human and machine interaction and holds significant application promise in the field of smart medical and wearable devices.

## 2. Materials and Methods

### 2.1. Materials

Polymide films with a thickness of 0.2 mm were purchased from Du Pont China Holding Co., Ltd., Shenzhen, China. A double-sided tap was purchased from Deli Group Co., Ltd., Ningbo, China. PDMS solution was purchased from Dow Corning and was prepared by mixing SYLGARD 184 and the hardener solution in a ratio of 10:1.

### 2.2. Instruments

A Huaray laser (Cypress2-355-15AY) was used for the fabrication of LIG. The laser power meter was purchased from Beijing Research Bond Technology Co., Beijing, China. A vacuum chamber (DZF-6050) was purchased from Hefei Kejing Material Technology Co., Ltd., Hefei, China. A high-temperature chamber (model: DKN612C) was purchased from Chongqing Yarmatec Co., Ltd., Chongqing, China. The sheet resistance measurement of the LIG was carried out using the SZT-C fast constant voltage test bench from JingGe Electronics Co., Ltd., Suzhou, China. Surface images and elemental distributions of the LIG were obtained using a tungsten scanning electron microscope and EDS (VEGA-4, TESCAN, Brno, Czechia). Raman spectra were studied using a laser micro-Raman spectrometer (Thermo Fisher Scientific, Waltham, MA, USA, DXR3, 532 nm). The change in the resistance of the flexible sensor was read using an ARDUINO Leonardo development board. Stretching of the flexible sensor was performed using a testing machine (EDT204A, Shenzhen Wance Testing Machine Co., Ltd., Shenzhen, China, EDT204A). The control parameters were cosine wave control with a displacement control mode, a period of 50, and a start speed and end speed of 0.05 mm/s.

### 2.3. Fabrication of Laser-Induced Graphene on PI Film

Figure 1 schematically illustrates the fabrication process of LIG. First, the PI film was cleaned with an ultrasonic cleaner and then adhered to the glass plate with PI double-sided tape to maintain its flatness after being exposed to the laser. The prepared PI sample was placed on the platform where the laser beam was focused. The laser beam was emitted by a 355 nm UV laser machine controlled by computer software. The output power was adjusted by a laser power regulator, the scanning path of the laser beam was controlled by a scanning mirror, and finally, the laser beam was focused on the PI surface by a field lens. The laser power was measured by a power meter.

### 2.4. One-Step Fabrication of Flexible Sensors

Figure 2 shows the one-step fabrication process of the flexible sensor based on the as-prepared LIG samples. The LIG samples were prepared as a 5 mm × 40 mm rectangular pattern under a laser power of 0.9 W, a scanning speed of 50 mm/s, a processing number of 10, and a scanning interval of 0.02 mm. The PDMS solution was poured into a mold with dimensions of 6 mm × 50 mm × 2 mm and degassed in a vacuum oven for over 60 min. At the same time, copper electrodes were connected to both ends of the LIG samples with conductive silver paste, and then the as-prepared samples were cured in a constant temperature oven at 100 °C for 60 min. The LIG samples were then placed on the PDMS solution and put back into the constant temperature oven at 100 °C for 90 min. To ensure that the LIG samples and the PDMS were combined tightly, a 500 g weight was placed on the top, as illustrated in Figure 2. After the PDMS solution was fully cured, the samples were taken out, and the excess part was cut off. The fabrication of the flexible sensors was finally completed.

## 3. Results and Discussion

### 3.1. Characterization of Laser-Induced Graphene

LIG was successfully fabricated by laser scribing at a laser power of 0.9 W, a scanning speed of 50 mm/s, a processing number of 10, a scanning interval of 0.02 mm, and an out-of-focus of 0 mm. Figure 3a depicts the scanning electron microscopy (SEM) image of the scale-like graphene on the PI sheet. The Raman spectra of the LIG showed three characteristic peaks of graphene (Figure 3b). The D peak at ~1350 cm^−1^ is a disordered vibrational peak of graphene due to amorphous carbon, defects and edges, etc. The G peak at ~1594 cm^−1^ represents the interaction between the carbon atoms in graphene and is used to detect the presence of graphene [44]. The G’ peak at ~2680 cm^−1^, which is also referred to as the 2D peak [45], represents the double resonance mode of graphene layers; it is used to describe the way carbon atoms stack up between layers in graphene samples and can be used to determine the layers and morphology of graphene [46,47]. The intensity ratios of I_D_/I_G_ were used to evaluate the defect density in the LIG, with a higher intensity ratio of I_D_/I_G_ reflecting more defects in the LIG sample [48]. The intensity ratios of I_2D_/I_G_ were used to evaluate the layers of graphene, with a higher ratio of I_2D_/I_G_ indicating fewer layers in the LIG [49].

The energy dispersive X-ray spectroscopy (EDS) results showed that the carbon content of the as-prepared sample was up to 96.16 wt.% and the oxygen and nitrogen contents were quite low [50] (Figure 3c–f).

### 3.2. Effects of Laser Parameters on Laser-Induced Graphene

#### 3.2.1. Effects of Laser Power on Laser-Induced Graphene

Figure 4a shows the Raman spectra of the LIG fabricated on a commercial PI sheet using different laser powers with the other laser parameters unchanged (scanning speed of 50 mm/s, processing number of 10, and scanning interval of 0.02 mm). It is obvious that LIG was successfully fabricated under all laser powers, as all three prominent peaks existed under all laser powers. The I_D_/I_G_ intensity ratio, as shown in Figure 4b, first decreased and then increased with an increase in laser power, and its minimum was observed at a laser power of 0.9 W. As indicated above, the lower the I_D_/I_G_ intensity ratio, the fewer the defects in the LIG sample. This means that the LIG sample with the fewest defects and a relatively better quality was obtained with a laser power of 0.9 W. Furthermore, the sheet resistances of LIG samples fabricated under different laser powers are exhibited in Figure 4c. The smallest sheet resistance of the as-prepared LIG samples (as low as 4.5 Ω per square) was also obtained under a laser power of 0.9 W. Comprehensively analyzing the results of the I_D_/I_G_ intensity ratios and the sheet resistances of the as-prepared LIG samples, the optimal LIG sample can be acquired at a laser power of 0.9 W.

The surface morphology of the LIG sample at a lower laser power of 0.3 W is shown in Appendix A. The high temperature generated by the laser breaks the C-H, C=O, C-O, and C-N bonds in PI film [51], forming gaseous products, such as N_2_, O_2_, and CH_4_ [52]. The porous surface structure is caused by gas leakage during the laser scribing process [53,54]. By contrast, the surface morphology of the LIG sample at a higher laser power of 0.9 W is exhibited in Appendix A. Scale-like structures instead of porous structures were obtained at higher laser powers [55]. This is due to the fact that there was an excessive amount of energy burned off the edges of the pores, closing them off [56,57].

The surface morphology of LIG showed a large number of porous structures prepared at a 0.3 W laser power. The porous morphology began to change to flakes as the laser power increased. When the laser power was 0.9 W, the surface of the LIG was nearly fully converted to flake graphene, giving the highest quality of LIG. As the laser power continued to increase, the flake structure was destroyed by the high-energy laser, which decreased the LIG quality, as shown in Appendix A.

#### 3.2.2. Effects of Scanning Speed on Laser-Induced Graphene

Figure 5a shows the Raman spectra of LIG fabricated on commercial PI film using different scanning speeds with the other laser parameters unchanged (laser power of 0.9 W, processing number of 10, and scanning interval of 0.02 mm). Figure 5b,c show that the intensity ratios and the sheet resistances of the LIG sample I_D_/I_G_ first decreased and then increased as the scanning speed increased, and both reached a minimum at a scanning speed of 50 mm/s.

A situation similar to laser cutting occurred and the PI film was punctured if the laser scanning speed was too low, resulting in a long laser irradiation time. With an increase in scanning speed, the PI film first generated LIG under the irradiation of laser energy. When the laser energy was further irradiated on the surface of the LIG, it caused the excitation and movement of the electrons of the LIG, resulting in an electron shielding effect. The electron shielding effect was the best and the quality of the generated lamellar LIG was optimal at a scanning speed of 50 mm/s. The previously prepared LIG was destroyed, and the quality of the LIG deteriorated due to the weakening of the electron shielding effect as the scanning speed continued to increase, as shown in Appendix A.

#### 3.2.3. Effects of Processing Numbers on Laser-Induced Graphene

Figure 6a shows the Raman spectra of LIG fabricated on commercial PI film using different processing numbers with the other laser parameters unchanged (laser power of 0.9 W, scanning speed of 50 mm/s, and scanning interval of 0.02 mm). LIG was successfully fabricated with processing numbers ranging from 5 to 15, as all three characteristic peaks existed in the Raman spectra. The I_D_/I_G_ intensity ratio, as shown in Figure 6b, firstly decreased and then increased with increasing processing numbers, and it was minimal at a processing number of 10. Furthermore, the sheet resistances of LIG samples fabricated under different processing numbers are shown in Figure 6c. The smallest sheet resistance was also obtained at a processing number of 10. According to the comprehensive analysis of the results of the I_D_/I_G_ intensity ratio and the sheet resistance of the LIG samples, the optimal LIG can be fabricated at a processing number of 10. The quality of the laser-induced graphene improved as the number of processes increased from 5 to 10 due to the removal of impurities and defects in the graphene. When the number of processes was more than 10, the repetitive scanning of the laser caused high-temperature damage to the LIG structure, leading to a reduction in LIG quality [58,59].

The SEM images of laser parameters with different processing numbers are shown in Appendix A; the LIG surface morphology consisted of a large number of microforms that were not transformed into sheet graphene at a processing number of 5. Porous graphene was further converted into flake graphene as the processing number increased, and the structure and quality of the LIG improved, with the optimum quality of the LIG achieved at a processing number of 10. However, increasing the processing number would destroy the surface morphology because of the energy of repeated processing, thus reducing the quality of the LIG.

#### 3.2.4. Effects of Scanning Interval on Laser-Induced Graphene

The laser spot size affects the scanning interval of the laser. The laser spot diameter formula is:(1)$$D_{0}=\frac{4M^{2}\lambda f}{\pi D}$$
where *D*_0_ is the spot diameter at the focal point; *M*^2^ = 1.2 is the beam quality; *λ* = 355 nm is the laser wavelength; *f* = 160 mm is the focal length of the focusing lens; *D* = 4.5 mm is the diameter of the incident laser beam; and *D*_0_ is calculated as ~0.02 mm.

Figure 7a shows the Raman spectra of the LIG fabricated on commercial PI film using different scanning intervals with the other laser parameters unchanged (laser power of 0.9 W, scanning speed of 50 mm/s, and a processing number of 10). The scanning interval directly affected the area and coverage of the laser action on the graphene surface, as well as the area D of laser damage and thermal effect on the graphene. Figure 7b,c show that the I_D_/I_G_ intensity ratios and the sheet resistances of the LIG samples initially decreased and then increased with an increasing scanning interval. The fragmentation and cracking of the graphene surface, affecting the adhesion of the graphene layer to the substrate, was due to the multiple overlay effect induced by laser irradiation at 0.01 mm. When the scanning interval was 0.03 mm, the PI film could not be completely covered during laser irradiation, so the PI film could not be completely converted into graphene, and the Raman spectra and sheet resistances verified that the quality of the LIG became poor [60] (Figure 7d).

Appendix A shows the SEM images of the LIG samples prepared with different scanning intervals. When the scanning interval was 0.01 mm, the pre-prepared LIG was ruined by the overlapped spots during the repeated process. At a scanning interval of 0.03 mm, there was a large amount of generated flake graphene on both sides of the processed spot with a PI film in between, which was not processed due to the oversized spot, affecting the quality of the prepared LIG.

### 3.3. Fabrication and Performance Tests of Flexible Sensors

#### 3.3.1. Performance Tests of the Flexible Sensors

Tensile and bending tests were performed to evaluate the performance of the prepared flexible sensors with a one-step method. The setup of the tensile test of the flexible sensors is shown in Figure 8a. The flexible sensor was fixed on the universal testing machine, and the ARDUINO microcontroller was connected to the copper electrode. The ARDUINO controller was used to read the voltage signals at the ends of the flexible sensors and convert the detected electrical signals into resistance outputs to the control panel. Cyclic tensile tests were applied to the flexible sensor. Figure 8b shows the response of the electrical resistance when the sensor was periodically stretched/released at the same strain (ε = 1%, 2%, 3%, and 4%). It is obvious that all sensors showed larger ∆R/R responses when larger tensile strains were applied. LIG is composed of a large number of graphene nanosheets, and due to the mechanical characteristics of this structure, when faced with various deformations due to stretching, bending, torsion, etc., a large number of microscale cracks are generated, resulting in changes in resistance [61]. In addition, even after the cycle test (50 iterations), the flexible sensors demonstrated no hysteresis, and *R*_0_ returned to the initial level during the relaxation time, indicating that the flexible sensors have good fatigue resistance and can be used in long-period applications.

Furthermore, the relationship between the relative change in resistance and the applied strain can be used to evaluate the performance of flexible sensors [62,63], which is indicated by the gauge factor (GF):(2)$$GF=\frac{\Delta R}{R_{0}\times \varepsilon }$$
where ∆*R* is the resistance change, *R*_0_ is the initial resistance of the flexible sensor, and ε is the applied strain. As illustrated in Figure 8c, the flexible sensor had a GF of up to 18.01 and an *R*^2^ of 0.99775, which means that the flexible sensor had a sensitivity in the strain range of 1% to 4%. The GF of this as-prepared flexible sensor was much higher than that in reference [37] (GF = 1.68) in the strain range of 1% to 10%.

Figure 8d shows the setup of the bending test of the flexible sensors. One end of the flexible sensor was fixed, and the degree of curvature is characterized by the bending angle, as illustrated in Figure 8d. The resistance was also recorded using an ARDUINO microcontroller during the bending process. The results (Figure 8e) showed a clear response of the sensor, with the resistance increasing as the bending angle increased (more bending). The increase in resistance can be interpreted according to the same explanation as the one given for the tensile tests: the bending results in the disconnection of the paper LIG fibers, which in turn leads to an increase in resistance. Furthermore, it is noteworthy that the response times of the sensors were 160 and 140 ms for the rise and fall times, respectively (Figure 8f).

More specifically, the LIG-based sensor also exhibited an excellent combination of quick response and exceptional durability. Figure 8g shows the response of relative resistance changes in the cyclic bending tests. After 1000 cycles of bending tests under a bending angle of 180 degrees, the response of the relative resistance changes in the flexible sensor was almost identical to that in the very first bending cycles (as illustrated in the zoomed-in section of the long-term tests, Figure 8g). This confirms its capability in wearable electronics.

#### 3.3.2. Application of the LIG-Based Flexible Sensors

With the advantages of in-time resistance response when bent and good flexibility, LIG-based sensors have the potential to be used to measure and track the movement of joints or objects. When integrated into gloves or other wearable devices, these sensors can accurately capture and translate hand and finger movements into digital signals, enabling precise control over virtual objects or computer interfaces.

Here, in order to demonstrate the applications of the LIG-based flexible sensors, a wearable glove was fabricated by adhering flexible sensors to the knuckles of a flexible glove. The bending of the fingers, which in turn led to resistance changes in the flexible sensors, was recognized by the ARDUINO microcontroller (Figure 9a). Figure 9b presents the real-time resistance variation of the LIG-based flexible sensors when different fingers were bent. The rising edge of a particular pulse indicates the bending motion of the corresponding finger, and the time duration of the sheet wave presents the duration of the bending motion of the finger. Moreover, the falling edge of the pulse indicates the straightening of the corresponding finger. The in-time and steady responses reveal the reliable recognition of finger bending motions of the LIG-based wearable HCI glove.

It was obvious that the wearable glove fabricated through the integration of LIG-based flexible sensors and a flexible glove could precisely detect the bending motion of different fingers. Furthermore, the application of the as-prepared wearable glove in HCI systems was explored. Figure 10a shows the basic setup of the HCI system. The LIG-based wearable glove was connected to the ARDUINO controller through an external circuit and then connected to the computer using a USB cable. The Mouse library provided by the ARDUINO IDE was used to control the mouse using the electrical signals from the flexible sensors as the variables. Flexible sensor 1 was set to control the left click of the mouse, flexible sensor 2 controlled the left movement of the mouse, flexible sensor 3 controlled the right movement of the mouse, flexible sensor 4 controlled the upward movement of the mouse, and flexible sensor 5 controlled the downward movement of the mouse. The bending motion of one particular finger resulted in a resistance change in the corresponding flexible sensor and was then detected by the ARDUINO controller. Then the ARDUINO controller executed the corresponding mouse movement function to achieve the corresponding mouse movement. As shown in the video (Appendix A), the wearer could easily type “HELLO” through the LIG-based HCI system on the computer screen. This demonstrates the potential of the LIG-based HCI system to be used by persons with limited mobility to control a virtual keyboard or mouse pointer, thus enabling them to perform tasks like typing, browsing the internet, and using software applications. This can significantly enhance their accessibility and independence in the digital realm.

## 4. Conclusions

In summary, LIG-based flexible sensors were successfully fabricated through a one-step method and applied for flexible HCI systems. The effect of laser parameters on the generated LIG was investigated and LIG samples with a sheet resistance of 4.5 Ω per square were acquired under the optimal laser processing parameters. The LIG-based flexible sensors showed an in-time response when bent, with a response time of 160 ms and a recovery time of 140 ms. Furthermore, the flexible sensors exhibited exceptional durability. After 1000 cycles of bending tests, the flexible sensors were structurally sound. The applications of the flexible sensors were demonstrated by a newly designed HCI system, and the usage of the HCI system was presented by typing words on a computer screen through the bending motion of human fingers. This paper offers a newly designed and cost-effective LIG-based flexible HCI system that demonstrates feasibility and independence in the digital domain. The device will enable people with limited mobility to carry out daily communication, and it will further expand the research on LIG applications, such as doctor–patient communication, HCI, and other applications.

## Figures and Tables

**Figure 1 materials-16-06938-f001:**
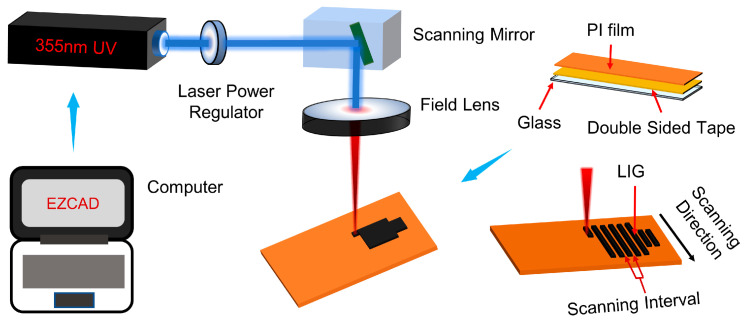
Schematic illustration of the LIG fabrication process.

**Figure 2 materials-16-06938-f002:**
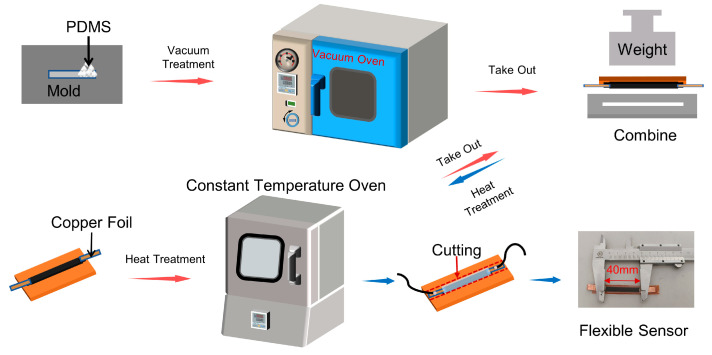
Schematic illustration of the one-step fabrication process of the LIG-based flexible sensors.

**Figure 3 materials-16-06938-f003:**
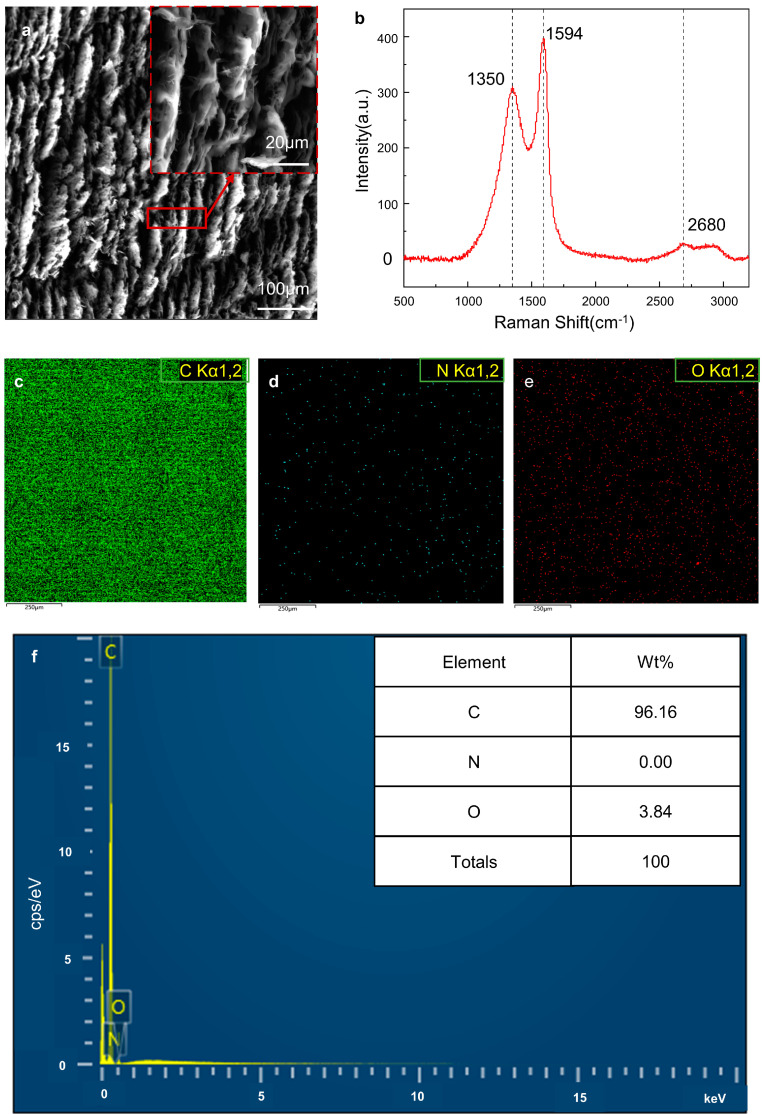
The morphology and characterizations of LIG under laser power of 0.9 W, scanning speed of 50 mm/s, processing number of 10, and scanning interval of 0.02 mm. (**a**). SEM images (1000× and 6000×) of LIG. (**b**). Representative Raman spectrum of LIG. (**c**). C element distribution on the LIG surface by EDS. (**d**). N element distribution on the LIG surface by EDS. (**e**). O element distribution on the LIG surface by EDS. (**f**). EDS elemental analysis of resulting LIG.

**Figure 4 materials-16-06938-f004:**
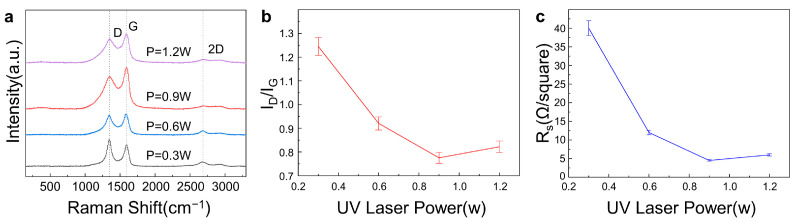
Effects of the laser power on the LIG quality with the scanning speed of 50 mm/s, the processing number of 10, and the scanning interval of 0.02 mm unchanged. (**a**). Raman spectroscopy of LIG at a different laser power. (**b**). Intensity ratio of the I_D_ peak to the I_G_ peak of LIG at different laser powers. (**c**). Sheet resistance (Rs) of LIG at different laser powers.

**Figure 5 materials-16-06938-f005:**
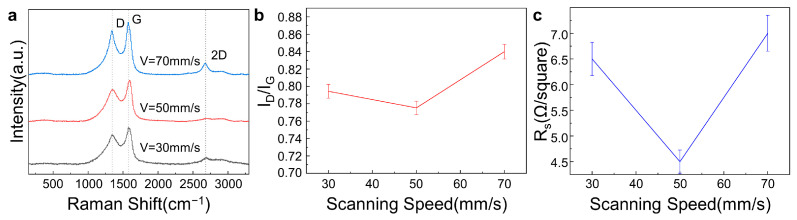
Effects of the scanning speed on the fabricated LIG with the laser power of 0.9 W, the processing number of 10, and the scanning interval of 0.02 mm unchanged. (**a**). Raman spectroscopy of LIG at different scanning speeds. (**b**). Intensity ratio of the I_D_ peak to the I_G_ peak of LIG at different scanning speeds. (**c**). Sheet resistance of LIG at different scanning speeds.

**Figure 6 materials-16-06938-f006:**
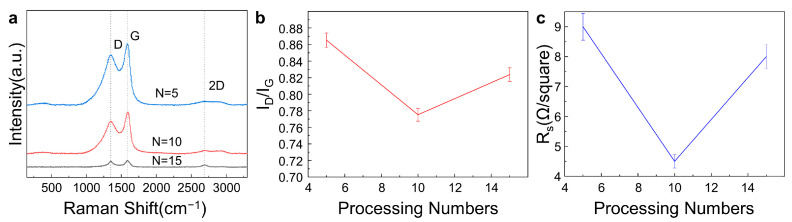
Effects of the number of processes on the LIG quality with the laser power of 0.9 W, the scanning speed of 50 mm/s, and the scanning interval of 0.02 mm unchanged. (**a**). Raman spectroscopy of LIG with different processing numbers. (**b**). Intensity ratio of the I_D_ peak to the I_G_ peak of LIG at different processing numbers. (**c**). Sheet resistance of LIG at different processing numbers.

**Figure 7 materials-16-06938-f007:**
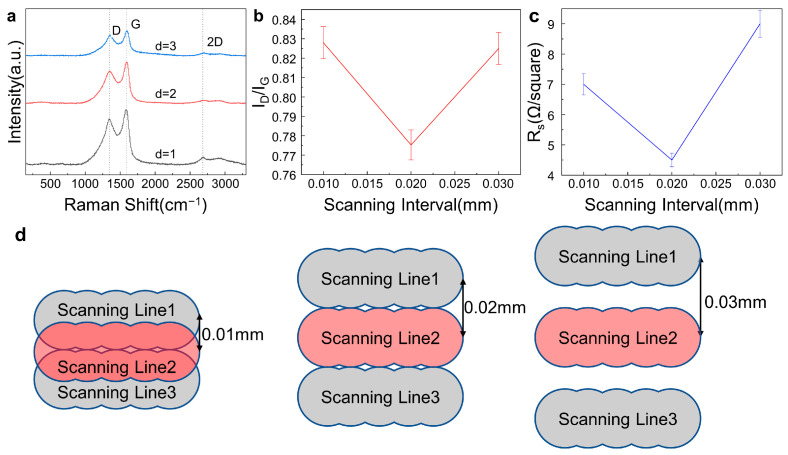
Effects of the scanning interval on the LIG characterization of LIG quality with the laser power 0.9 W, the scanning speed of 50 mm/s, and the processing number of 10 unchanged and a schematic diagram of processing by varying the scanning interval. (**a**). Raman spectroscopy of LIG at different scanning intervals. (**b**). Intensity ratio of the I_D_ peak to the I_G_ peak of LIG at different scanning intervals. (**c**). Sheet resistance of LIG at different scanning intervals. (**d**). Schematic diagram of scanning at different scanning intervals during laser fabrication.

**Figure 8 materials-16-06938-f008:**
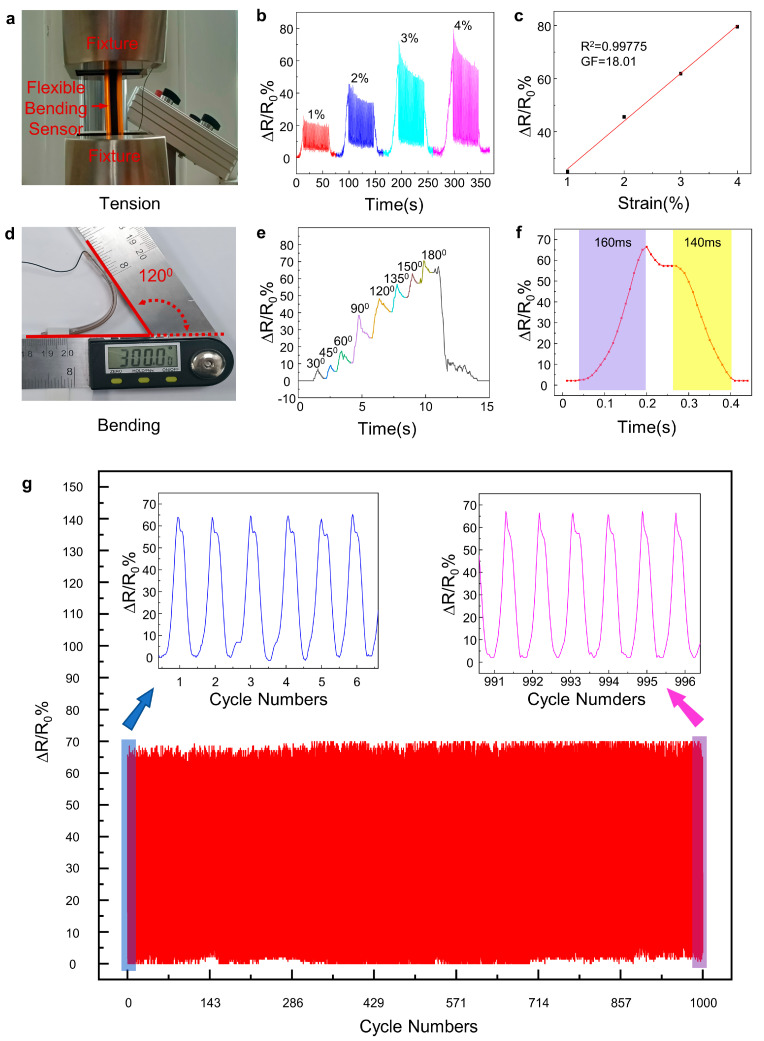
Performance of the flexible sensor. (**a**). The flexible sensor was clamped under a tension device to test the performance. (**b**). Fifty tension cycles at 1%, 2%, 3%, and 4% strain on a testing machine. (**c**). Variation of resistance with strain, indicating strain sensitivity. (**d**). Bending test image of the flexible sensor. (**e**). Flexible sensor resistance variation at different bending angles. (**f**). Response time and recovery time of the flexible sensor under the 180-degree bending condition. (**g**). Cycle test at a bend of 180 degrees.

**Figure 9 materials-16-06938-f009:**
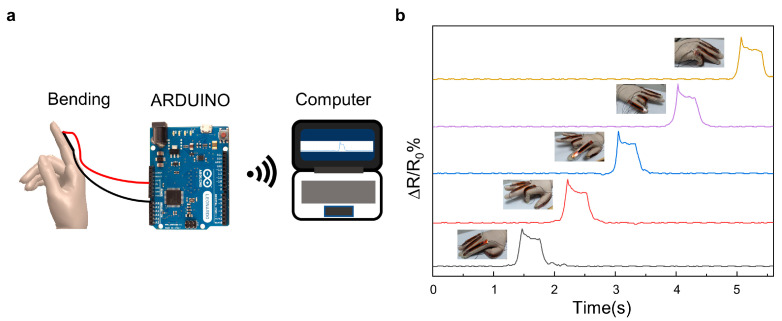
Real-time feedback from the flexible glove’s per-finger flexion tests. (**a**). Schematic illustration of the flexible sensors. (**b**). Bending testing response per finger.

**Figure 10 materials-16-06938-f010:**
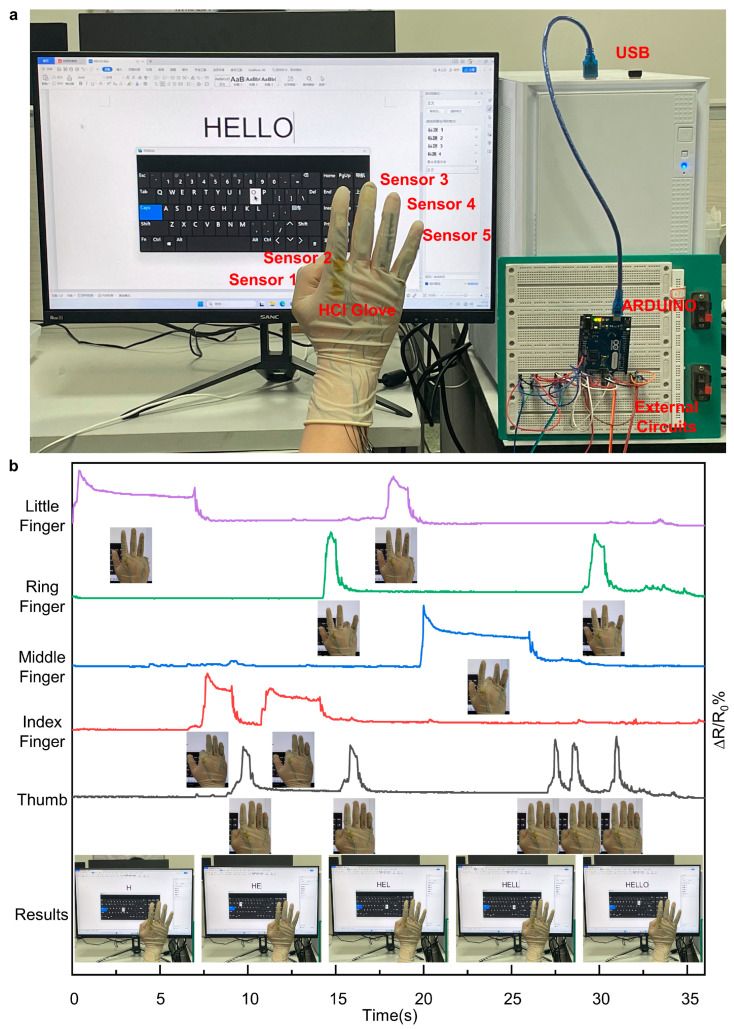
Basic setup and signal detection of the HCI system. (**a**). The basic setup of the HCI system. (**b**). Flexible sensors detect when typing “HELLO” on the computer.

## Data Availability

Data are contained within the article or Appendix A or are available upon request from the corresponding authors.

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
