# Peer review of "Fabrication of Laser-Induced Graphene Based Flexible Sensors Using 355 nm Ultraviolet Laser and Their Application in Human–Computer Interaction System"

_materials, 2023, doi:10.3390/ma16216938_

Round 1

Reviewer 1 Report

Comments and Suggestions for Authors

The manuscript "Fabrication of Laser-induced Graphene Based Flexible Sensors 2 Using 355 nm Ultraviolet Laser and Their Application in Human-Computer Interaction System" covers the creation of a new flexible sensor based on LIG obtained from Polyimide using a 355 nm UV laser. It describes how several parameters, such as laser power, scanning speed, processing numbers, and scanning intervals, affect the quality of the LIG produced. The manuscript is well-structured, and the content is relevant to the researchers working on the topic. As such, I think that this manuscript is interesting, however authors should consider the following points to improve the quality.

Comments:

1) Add the brand and reference of double-sided tape and polymers to the materials and methods.

2) Make it clearer in the abstract that LIG is generated through direct laser interaction with Polyimide.

Comments on the Quality of English Language

English could be improved specially in the last paragraph of the introduction.

Author Response

We thank you very much for your helpful comments and patience on our manuscript entitled “Fabrication of Laser-induced Graphene Based Flexible Sensors Using 355 nm Ultraviolet Laser and Their Application in Hu-man-Computer Interaction System” (Ref. No.: materials-2676562). Now we have revised the manuscript according to your guidance and comments.

We have studied the reviewers’ comments carefully and have made revisions which marked in blue in the paper. We have tried our best to revise our manuscript according to the comments. Attached please find the revised version, which we would like to submit for your kind consideration.

Thanks very much for your attention to our paper and your great help to our paper processing.

(1) Add the brand and reference of double-sided tape and polymers to the materials and methods.

We have added the brand and reference of double-sided tape and polymers to the materials and methods in the article. The relative description was added in the revised manuscript (Paragraph 1, Page 3).

(2) Make it clearer in the abstract that LIG is generated through direct laser interaction with Polyimide.

We have revised the abstract according to the reviewer’s comment. It is clear that LIG is generated through direct laser interaction with Polyimide in the abstract.

(3) Comments on the Quality of English Language: English could be improved specially in the last paragraph of the introduction.

We have checked our manuscript and refine the language of our manuscript thoroughly, especially the last paragraph of the introduction.

Reviewer 2 Report

Comments and Suggestions for Authors

The authors described flexible sensors prepared by fast laser irradiation and their application in human-computer interface. It may be considered as a promising step towards development of wearable electronics; however, it should be introduced in more details what has been achieved in this area. In the introduction, it is stated that flexible sensors based on laser-induced graphene has been already introduced. Their applications and properties should be given in more details to show what the novelty of the reviewed manuscript is.  

Other comments, question:

Furthermore, more details should be provided about the reading of the sensors signal – it is only stated that it was achieved using Arduino controller, but it is worthy to give more details about it.

The text also need formal revision to correct typos and mistakes (e.g. Fig. 8 on page 3 – should be Fig. 2; missing caption of figure on the page 3; missing verb in the first sentence of the 2.2 section and so on.)

How reliable are last two decimal numbers of the achieved GF value? Furthermore, the achieved properties of the sensor should be compared with properties of other flexible resistive strain sensors referred to in the Introduction section.

Comments on the Quality of English Language

The English is comprehensive, but editions are required (e.g. "...morphology exists large number of structures..."; "...which can be fabricated the optimal LIG..." and so on) - see also Comments and Suggestions for Authors

Author Response

We thank you very much for your helpful comments and patience on our manuscript entitled “Fabrication of Laser-induced Graphene Based Flexible Sensors Using 355 nm Ultraviolet Laser and Their Application in Hu-man-Computer Interaction System” (Ref. No.: materials-2676562). Now we have revised the manuscript according to your guidance and comments.

We have studied the reviewers’ comments carefully and have made revisions which marked in blue in the paper. We have tried our best to revise our manuscript according to the comments. Attached please find the revised version, which we would like to submit for your kind consideration.

Thanks very much for your attention to our paper and your great help to our paper processing.

(1) The authors described flexible sensors prepared by fast laser irradiation and their application in human-computer interface. It may be considered as a promising step towards development of wearable electronics; however, it should be introduced in more details what has been achieved in this area. In the introduction, it is stated that flexible sensors based on laser-induced graphene has been already introduced. Their applications and properties should be given in more details to show what the novelty of the reviewed manuscript is.

We have added the properties of the flexible sensors based on laser-induced graphene and their applications in the references we cited in the introduction. The relative description was added in the revised manuscript (Paragraph 3, Page 2).

(2) Furthermore, more details should be provided about the reading of the sensors signal – it is only stated that it was achieved using Arduino controller, but it is worthy to give more details about it.

Thanks to the reviewer’s comments. More details of the reading process of the sensors signal have been added in the revised manuscript (the last paragraph in Page 8).

(3) The text also need formal revision to correct typos and mistakes (e.g. Fig. 8 on page 3 – should be Fig. 2; missing caption of figure on the page 3; missing verb in the first sentence of the 2.2 section and so on.)

We have checked our manuscript and corrected the typos and mistakes according to the reviewer’s comments. The revised texts are marked blue in the revised manuscript.

(4) How reliable are last two decimal numbers of the achieved GF value? Furthermore, the achieved properties of the sensor should be compared with properties of other flexible resistive strain sensors referred to in the Introduction section.

We are sorry about the confusion of the achieved GF value. The GF value presented in the manuscript was obtained through the calculation and noted rounded. We have rounded the GF to18.01 according to the reviewer comments.

Furthermore, the achieved properties of the sensor were compared with properties of other flexible resistive strain sensors referred to in the Introduction section according to the reviewer’s comments and the relative descriptions were added in the revised manuscript.

(5) Comments on the Quality of English Language: The English is comprehensive, but editions are required (e.g. "...morphology exists large number of structures..."; "...which can be fabricated the optimal LIG..." and so on) - see also Comments and Suggestions for Authors

We have checked our manuscript and corrected the typos and mistakes according to the reviewer’s comments. The revised texts are marked blue in the revised manuscript.

Reviewer 3 Report

Comments and Suggestions for Authors

The paper is well-written and contains some useful data. It can be accepted after addressing the following:

The authors need to further elaborate on the properties of graphene and explain why it is important for this particular application in the introduction.

They should also elaborate more on how the findings of this study would enhance the existing literature and current knowledge.

Comments on the Quality of English Language

The paper is well-written and contains some useful data. It can be accepted after addressing the following:

The authors need to further elaborate on the properties of graphene and explain why it is important for this particular application in the introduction.

They should also elaborate more on how the findings of this study would enhance the existing literature and current knowledge.

Author Response

We thank you very much for your helpful comments and patience on our manuscript entitled “Fabrication of Laser-induced Graphene Based Flexible Sensors Using 355 nm Ultraviolet Laser and Their Application in Hu-man-Computer Interaction System” (Ref. No.: materials-2676562). Now we have revised the manuscript according to your guidance and comments.

We have studied the reviewers’ comments carefully and have made revisions which marked in blue in the paper. We have tried our best to revise our manuscript according to the comments. Attached please find the revised version, which we would like to submit for your kind consideration.

Thanks very much for your attention to our paper and your great help to our paper processing.

(1) The authors need to further elaborate on the properties of graphene and explain why it is important for this particular application in the introduction.

we have revised the introduction of the manuscript and added more detailed properties of the graphene, which would explain the advantages of graphene in the applications of flexible sensors. The relative descriptions were added and marked blue in the revised manuscript.

(2) They should also elaborate more on how the findings of this study would enhance the existing literature and current knowledge.

We appreciated the reviewer’s suggestions and added relative context in the revised manuscript to state how the findings of this study would enhance the existing literature and current knowledge.